# Online Teaching Alternative in Human Anatomy

**Alberto Garcia Barrios** [1,2], **Ana Isabel Cisneros Gimeno** [1,2,*], **María Camen Garza García** [1,2], **Itziar Lamiquiz Moneo** [1,3] and **Jaime Whyte Orozco** [1,2]

1. Department of Human Anatomy and Histology, Faculty of Medicine, University of Zaragoza, 50009 Zaragoza, Spain; agarciab@unizar.es (A.G.B.); mcgarza@unizar.es (M.C.G.G.); itziarlamiquiz@unizar.es (I.L.M.); jwhyte@unizar.es (J.W.O.)
2. Medical and Genetic Research Group (GIIS099) Aragon Health Research Institute (IIS Aragón), 50009 Zaragoza, Spain
3. Miguel Servet University Hospital (GIIS007) Aragon Health Research Institute (IIS Aragón), CIBERCV, 50009 Zaragoza, Spain
* Correspondence: aicisner@unizar.es

**Abstract:** The objective was to implement a "Breakout" activity using an online platform (Genially®) similar to those carried out in person to increase interactivity, motivation, and teamwork. The activity was proposed during the 2021–2022 academic year in the Human Anatomy II (Splanchnology) course taken in the second semester of the Bachelor's Degree in Medicine (University of Zaragoza, Zaragoza, Spain) and was carried out with the participation of 89 students enrolled in the course. The evaluation of the experience by the students was carried out by means of an online questionnaire that comprised four questions (based on the Likert scale) and by the teaching staff through a coordination meeting. In total, 86% of the students agreed regarding the positive effect of these kinds of activities on interactivity and motivation, with 65% agreeing on the usefulness of these tools. Around 70% agree that this activity helped them to integrate course content and to enhance teamwork.

**Keywords:** human anatomy; splanchnology; innovation; breakout

## 1. Introduction

The reform of educational models promoted by the European Higher Education Area (1999) has raised the need to adapt teaching to the needs of today's working world and to focus on students as critical but above all participatory and active subjects [1]. For this, teachers must make an effort to innovate the methodology and tools that have been used in teaching until now to meet the teaching competencies set out in the guide for each subject [2].

Currently, the profile of students in the classroom is much more technological than previous generations, and this forces teachers to carry out activities that complement and/or replace the "master classes" that have been used until now, with the aim of increasing or at least maintaining the attention, motivation, and participation of students [3].

Gamification, also known as Game-Based Learning (GBL) or game-based learning (ABJ), has been proposed as a teaching alternative in which games are implemented in the educational environment at all levels to improve the teaching–learning process [4,5].

There are numerous activities that can be considered as gamification, and Escape Games, Escape Rooms, or "escape rooms" are widely used in innovative teaching and encourage interaction between students to achieve the goal of escaping from the room in which they are locked in within a limited amount of time. Hall Escape Room or Breakout activities present a similar dynamic in terms of their resolution but do not pose the need to escape and are different from each other. In the former, the activity is carried out in a physical "room", while in the latter, the activity is carried out entirely online [6,7].

The objective in this work was to implement a Breakout activity using online platforms (Genially®) that was similar to those carried out in person to promote motivation and to

improve the teaching–learning process and group work among the students in the Human Anatomy II (Splanchnology) course implemented in the Bachelor's Degree in Medcine (University of Zaragoza, Zaragoza, Spain) and its subsequent evaluation by the students.

## 2. Materials and Methods

A Breakout activity was planned for one of the practical sessions of the 2021–2022 academic year of the Human Anatomy II (Splanchnology) course, with a total of 89 students enrolled and taken during the third semester of the Bachelor's Degree in Medicine. For this purpose, and for management and capacity control reasons, the students were divided into subgroups of four students who had to work together to achieve the set objective.

To carry out this activity, which was carried out entirely on electronic devices, class instructors designed a Breakout activity on the online platform Genially®. The design of the activity was based on "getting out" of the Faculty of Medicine after a closure caused by a bacteriological leak in one of its laboratories. To achieve the objective, students had to solve a series of riddles and overcome challenges whose common thread was the theoretical–practical syllabus of the course.

The activity, which was set in a "virtual classroom" similar to the real one, began with an explanatory video in which they were informed that the only way to "escape" was to reach the dissection room and to unlock the exit door with a numerical code.

To do this, they had to look for the information necessary to decipher the code that would allow them to leave the room, the door of which had been blocked, and advance to the entrance of the Human Anatomy Department, where they were received by the secretary of the department. Once there, they were informed of the need to solve a series of clinical cases associated with the syllabus in order to be given a key to gain access to the dissection room.

Once they reached the dissection room, they were assigned four consecutive challenges based on the practical portion of the course, the resolution of which provided them with a numerical digit that they had to remember in order to decipher the password that allowed them to leave the centre.

The team that managed to "escape" first was rewarded with 0.1 extra points towards their final mark in the course.

The evaluation of student satisfaction was carried out on a voluntary basis by means of an online questionnaire comprising questions (Table 1) based on a Likert scale (with five response options, with 1 representing totally disagree and 5 representing totally agree).

**Table 1.** Questionnaire for assessing the activity proposed to students.

| | |
|---|---|
| Question 1 | Do you think that the use of Escape Room activities on an electronic device motivates the student and makes the practical sessions more interactive? |
| Question 2 | Do you think that Breakout gamification activities on electronic devices are a useful teaching method for learning? |
| Question 3 | Do you think the activity has helped you to integrate theory and practice? |
| Question 4 | Do you think Escape Room activities encourage teamwork to achieve the final objective? |

The assessment by the teaching staff was carried out by means of a coordination meeting.

## 3. Results

The evaluation survey, which was voluntary, was filled out by 88% (78/89) of the students who participated in the activity. From their answers, the following data were obtained:

The first question (Figure 1) asked about the interactive and motivational effect of these activities in the practical sessions, where 86% of the student body agreed (28%) or strongly agreed (58%) on the positive effect of these types of activities on interactivity and motivation, while the remaining 14% were neutral.

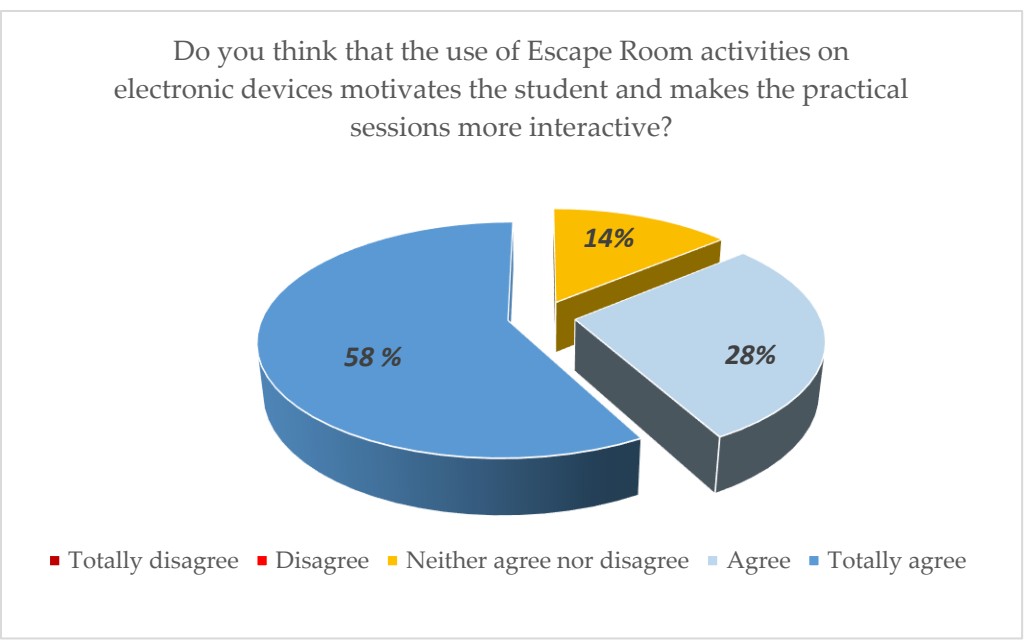

**Figure 1.** Assessment of the activity's effect on motivation and interactivity.

Figure 2 shows the results of the second question of the questionnaire, which asks about the usefulness of these activities for learning. In it, it can be seen that 65% of the answers agree (28%) or totally agree (37%) regarding the usefulness of these tools, while the remaining 35% do not consider them useful but do not rule them out as teaching tools either.

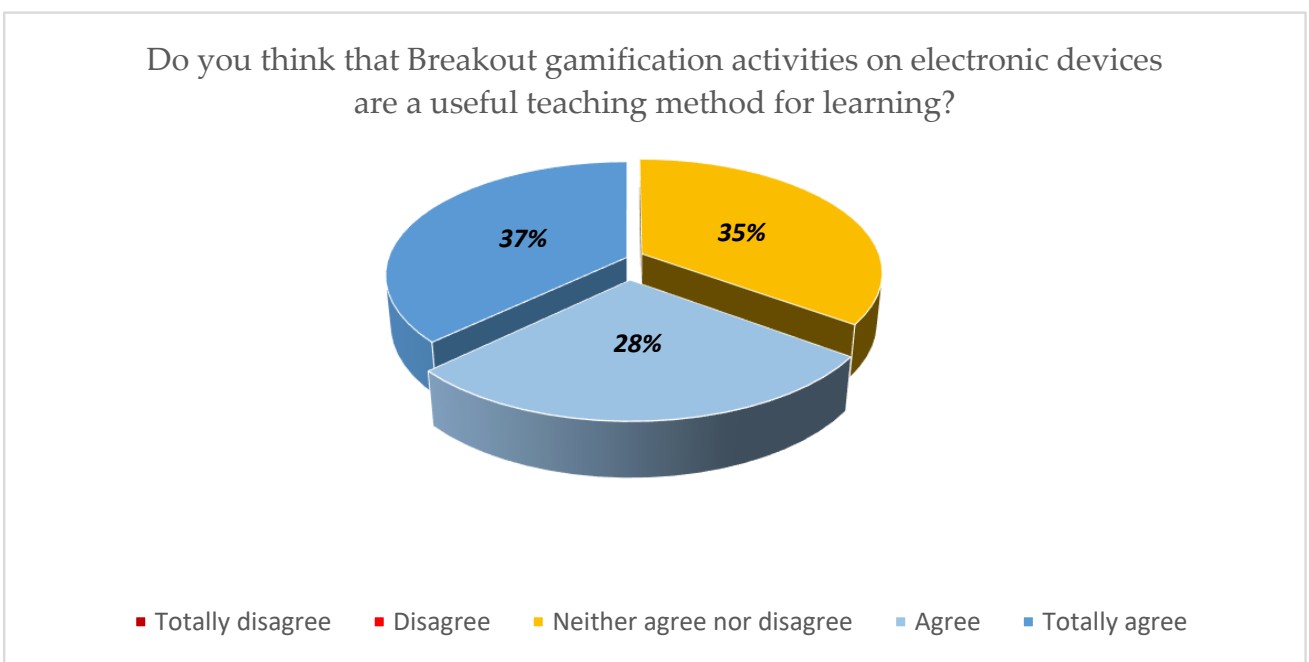

**Figure 2.** The usefulness of these activities for learning.

In the third and fourth questions of the questionnaire (Table 2), the students' opinions on the capacity of this type of activities to integrate the theoretical and practical contents of the subject was determined as well as the capacity of these activities to promote teamwork among students. In the survey, 72% of the respondents agreed (28%) or strongly agreed (44%) that this activity has helped them to integrate the course content, while 28% were

neither in favour nor against the usefulness of the activity. On the other hand, 79% of the responses affirm that these activities enhance teamwork, while the remaining 21% are neutral to this premise.

**Table 2.** Evaluation of the effect of the activity in terms of integrating course content and promoting teamwork.

| | Totally Agree | Agree | Neither Agree or Disagree | Disagree | Totally Disagree |
|---|---|---|---|---|---|
| Do you think the activity has helped you to integrate theory and practice? | 44% | 28% | 28% | 0% | 0% |
| Do you think the activity enhances teamwork to achieve the final goal? | 58% | 21% | 21% | 0% | 0% |

## 4. Discussion

New teaching methodologies, including gamification and game-based learning, can be considered as tools that complement and replace the theoretical–practical sessions of a course while also encouraging student participation and motivation. In this area, as corroborated in our study, activities that bring games into the classroom generate a more favourable environment for students, encouraging participation and improving motivation towards the subject [8,9]. On the other hand, previous studies carried out by this teaching team have already shown how the use of games in the classroom is positive when it comes to motivating students in the teaching–learning process for human anatomy [2] and for positively reinforcing those who solve the riddles more quickly [10].

However, despite the proven positive effect of this type of methodology, its use among teachers of previous generations is difficult, either because of difficulties encountered when handling digital technology or fear of "change", and therefore, the master class model, in which students become passive subjects, is maintained, and their motivation and participation may be lower [11]. However, we believe that far from being eliminated, this type of teaching should be considered as a basis that facilitates the integration of basic concepts that will later be implemented in these activities.

## 5. Conclusions

GBL, and specifically digital Escape Room activities, can be considered as useful teaching tools that not only favour learning, but that also improve the integration of content and that also motivate and encourage teamwork among students.

**Author Contributions:** Conceptualization, A.G.B.; methodology, A.G.B.; data curation, A.G.B., M.C.G.G., I.L.M.; writing—original draft preparation, A.I.C.G.; writing—review and editing, A.I.C.G.; supervision, J.W.O. All authors have read and agreed to the published version of the manuscript.

**Funding:** This research received no external funding.

**Institutional Review Board Statement:** Not applicable.

**Informed Consent Statement:** Not applicable.

**Acknowledgments:** Evaluation of knowledge acquisition was not performed: an experiment evaluating knowledge acquisition after the intervention and comparing performance with a control group should be carried out. Link to activity: https://view.genial.ly/5fa798689bff5f0cee6d09a5/game-breakout-escape-room (accessed on 30 June 2022).

**Conflicts of Interest:** The authors declare no conflict of interest.

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
