# Peer review of "Online Teaching Alternative in Human Anatomy"

_2813-0545, doi:10.3390/anatomia1010009_

Round 1

Reviewer 1 Report

1. The manuscript is well-written and presents clearly and concisely a novel approach to increase interactivity, motivation and teamwork among the students when learning anatomy.

2. Although the proposal presented here seems to be original and powerful to increase motivation in the students, access to the resource is not provided. Without availability of the resource, the contribution of this work is scarce, and its reproducibility is unfeasible. I recommend providing free access to the developed learning tool. In this way, students from all over the world and academic community could benefit from the proposal, increasing the impact and contribution of the work.

3. The authors show that the implementation of a "Breakout" activity increase interactivity, motivation, and teamwork among the students. However, whether this approach improved learning has not been evaluated. An experiment evaluating knowledge acquisition after the intervention and comparing performance with a control group should be included. Otherwise, this might be reflected as a limitation of the study. 

4. Colors of the graphs are confusing, as similar colors are used for opposite concepts (i.e. agree – disagree, and totally agree - totally disagree). I suggest using the same range of colors for agree – totally agree (e.g. light blue for agree and dark blue for totally agree) and warmer colors for disagree (e.g. yellow disagree and red for totally disagree).

5. I suggest including more international references as most of them are in Spanish

6. Minor comments: in line 37 the term “escape rooms” is repeated

Author Response

Thank you very much for your comments. We appreciate the input you have provided on our manuscript. Following your suggestions, we have added the next modifications:

-We have added the corresponding link at the end of the technical note, so that the activity can be accessed and shared among the teachers who read the journal.

-As this activity did not consider the possibility of a subjective assessment of learning improvement, we have added this as a limitation of the study.

Reviewer 2 Report

I think the topic of this article is very intriguing, particularly for high-school A&P teachers! A&P teachers can have difficulty engaging their students in the learning content if they do not have access to cadavers for their courses, so this could be a great way to get their students engaged! 

Author Response

Thank you very much for your contribution to our manuscript and its effect on education, especially in higher education. The text of the technical note has been revised to improve the English spelling.

Round 2

Reviewer 1 Report

The authors have improved the manuscript after the revision process. I would like to highlight only two minor comments before final publication:

1. Authors change colors of the graph making them more visual and easier to understand. However, they removed from the legend the options “disagree” and “totally disagree”, I suggest adding them to the legend (in hot colors) although they have 0% rate response. This will add value to the proposal, as readers can see from the graphs that these options were available, but no one disagree with the potential of the tool. In the same way I would add these two columns in Table 2, including the five response options and all the results, also 0% rate response is a result and must be shown.

2. I would change the limitation of the study sentence by: “Evaluation of knowledge acquisition was not performed”

Author Response

Thank you very much for the last minor comments. We have already modified them in the text.

The problem is that I can't attach the file where this modification of the figures-tables is, because there is no option to do it.
